# Photonic Materials Cloud: An Online Interactive Open Tool for Creating, Comparing, and Testing Photonic Materials

**DOI:** 10.3390/nano12152585

**Published:** 2022-07-28

**Authors:** Matiyas Tsegay Korsa, Søren Petersen, Neda Rahmani, Alireza Shabani, Yogendra Kumar Mishra, Jost Adam

**Affiliations:** 1Computational Materials Group, SDU Centre for Photonics Engineering, Mads Clausen Institute, University of Southern Denmark, DK-5230 Odense, Denmark; matiyas@sdu.dk (M.T.K.); neda@sdu.dk (N.R.); shabani@sdu.dk (A.S.); 2Department of Mechanical and Electrical Engineering, University of Southern Denmark, DK-6400 Sønderborg, Denmark; sope@sdu.dk; 3Centre NanoSyd, Mads Clausen Institute, University of Southern Denmark, DK-6400 Sønderborg, Denmark; mishra@mci.sdu.dk

**Keywords:** plasmonics, engineering education, meta-material, optical constants, electromagnetic modeling, nanoparticles

## Abstract

Recent advances in nanoscale fabrication and characterization further accelerated research on photonics and plasmonics, which has already attracted long-standing interest. Alongside morphological constraints, phenomena in both fields highly depend on the materials’ optical properties, dimensions, and surroundings. Building up the required knowledge and experience to design next-generation photonic devices can be a complex task for novice and experienced researchers who intend to evaluate the impact of subtle material and morphology variations while setting up experiments or getting a general overview. Here, we introduce the Photonic Materials Cloud (PMCloud), a web-based, interactive open tool for designing and analyzing photonic materials. PMCloud allows identification of the subtle differences between optical material models generated from a database, experimental data input, and inline-generated materials from various analytical models. Furthermore, it provides a fully interactive interface to evaluate their performance in important fundamental (numerical) optical experiments. We demonstrate PMCloud’s applicability to state-of-the-art research questions, namely the comparison of the novel plasmonic materials aluminium-doped zinc oxide and zirconium nitride and the design of an optical, dielectric thin-film Bragg reflector. PMCloud opens a rapid, freely accessible path towards prototyping optical materials and simple fundamental devices and may serve as an educational platform for photonic materials research.

## 1. Introduction

Light–matter interaction is one of the most critical parts of our daily life, offering a wide range of exciting phenomena and applications, from fundamental Rayleigh scattering to sensing and imaging. Many systems are derived from it, including optoelectronic devices such as light-emitting diodes, solar cells, quantum devices, optical communication applications, and data transfer technology [1,2,3]. During the past few decades, researchers have made extensive efforts to control, confine, and guide the electromagnetic waves within devices by manipulating material properties or using geometric engineering to enhance the performance of optical devices. Theoretically, the light–matter interaction mechanisms entirely depend on the material, the geometry dimension, and the wavelength of incident light. Hence, one can divide them into two main subcategories from the viewpoint of solving the governing equation in the problem. First, classical (Maxwell equation-based) governing equations explain problems in the case of an incident wavelength greater than, or in the same order of, the smallest geometric features [4,5]. Second, quantum mechanical rules between atoms and molecules in atomic-scale interactions address the light–matter interactions based on the excitation of electrons between electronic and atomic orbitals [6]. In addition to the geometry, material features play a crucial role in light–matter interaction [7]. Metals have the capacity to interact with light through their free electrons by mimicking the electric field component of the incident beam and creating oscillating collective electrons at the metal–dielectric interface. This effect is the so-called surface plasmon phenomenon, one of the fascinating aspects of light–matter interaction and studied disciplines in optics, filling the gap between conventional optics and highly integrated nanophotonic components [8,9]. Recent developments in plasmonic research expanded its usage in emerging applications, including clocks, metamaterials, sensing, biomedicine, and plasmonically enhanced solar cells, and it is expected to keep growing as a result of the increasing capabilities with regards to fabrication and characterization of plasmonic nanoparticles [10,11,12,13,14,15]. Another vital material class to interact with light, considered an alternative for plasmonic, is the class of high-refractive-index dielectrics and semiconductors. These materials can absorb electromagnetic waves (in the case of semiconductors) above their bandgap, and the high refractive index helps confine and diffract the light if adequately engineered. Due to the low optical loss and not being confined to the surface compared to metals, they create more intense electric fields than metals limited to the surface and interface. They also possess the advantages of exciting the fundamental physical mechanism of light–matter interaction, including magnetic and electric dipole/multipoles where the metals suffer the lack of magnetic modes except in the specific geometries.

The emergence of advanced mathematics, numerical methods, and powerful computational resources have tremendously helped the community figure out light–matter interactions and perform detailed studies of their underlying physics without facing the severe struggles (limitations) of experimental processes. As mentioned before, a part of these numerical methods focuses on solving classical Maxwell’s equations in a light–matter problem, including material feature, geometry, and incident light properties. Among these methods, we can point out the well researched and popular ones such as Mie theory, which formulates light scattering of spherical particles [16,17], and the transfer matrix method (TMM) for modeling and optimizing the optical response of thin films, with the advantage of lower simulation times compared to its counterparts [18,19,20].

Continuous, intense research in nanoparticles and thin films is still ongoing due to their numerous applications in biomedical, solar cell, UV protection, coating, and energy storage fields. Different studies showed that analyzing the optical properties of either structure is essential for optimizing, modifying, and enhancing the optical response in corresponding applications. Here, we introduce our recently released online interactive tool *Photonic Materials Cloud* (PMCloud) [21], which covers the light–matter interaction in nanoparticles/thin films to a full extent. In the following sections, we will show how well our dashboard results can predict the optical response of different structures compared with experimental and theoretical results in the literature. Currently, some tools exist which help with the study of optical systems and their material properties. They include web pages such as the refractive index database [22], which helps to obtain experimental models of various materials’ optical properties represented in refractive indices. However, it is tedious to compare material responses, and one cannot design material models. Another tool is Scott Prahl’s online Mie scattering calculator [23], which allows for computation of the Mie scattering of dispersed particles in a medium. However, this tool only computes the scattering for one wavelength at a time, and the user has to manually input the wavelength and refractive index for each data point they wish to compute. PMCloud, for starters, combines and extends the advantages of the above two tools in a single online platform. Developed and maintained by the computational materials group developed at SDU, it integrates the material library and allows a materials list to be created for further handling. The user can add material models from their own CSV files, using Drude–Lorentz parameters, or simply a constant refractive index material by entering the real and imaginary (*n* and *k*) values. With the initially created materials list, the user can then compare selected materials via interactive, multi-material dispersion plots. The click of a button will download either the plot data (as a CSV file) or the corresponding image in a publication-ready format. The Mie scattering tab also improves Scott Prahl’s online Mie scattering calculator, since it computes the scattering in a wavelength range and discretization, which the user chooses. Similarly, for the thin-film optical response, PMCloud can assign layers of the defined material models, which again are plotted as 1D plots over wavelength or incidence angles, or as 2D plots.

PMCloud is designed as a freely accessible online tool. While pieces of PMCloud’s functionalities are certainly available, or can be established in (potentially commercial) software, e.g., COMSOL Multiphysics, Lumerical FDTD solutions, and others, the possibility of rapidly and interactively comparing and testing optical materials is, to the best of our knowledge, unique at this point.

The following sections exemplify PMCloud’s capabilities with typical use cases.

## 2. Use Cases—What can the PMCloud Help with?

PMCloud’s target user groups include experimentalists, who seek an easy-access platform to design or cross-validate their experimental setups and results. To this end, PMCloud firstly allows for the upload of experimental material data and its direct comparison to database or theory-based materials. Secondly, with the uploaded or generated data, the user can perform a series of standard scattering and thin-film optical experiments, predicting the optical response of a wide range of geometries, and thereby helping to validate or modify experiments in a time-efficient way (Figure 1). The next three sub-sections sketch typical use cases, which we will back up with specific examples in the numerical results section.

### 2.1. Plasmonic Bio-Sensing

The plasmonic response is a novel characteristic of nanostructures built from noble metals and metal-like materials. The plasmonic behavior is very sensitive to the nanostructures’ size and shape and the surrounding medium’s dielectric constant. Especially due to this sensitivity to the dielectric environment, if any external object (e.g., chemical, biological, nanostructure) interacts with a plasmonic/photonic platform, the plasmonic response will change as the reflection of this interaction. If quantified and understood carefully, this can lead to excellent and precise sensing platforms [24,25,26]. To this end, theoretically and experimentally, a detailed understanding of the nanostructures’ plasmonic behavior is critical. Often, experimental and theoretical research efforts move independently. Nevertheless, their synchronization via a common platform can help better understand and predict these materials’ plasmonic properties in a real sense, and it can open many new avenues in reliable sensing and photonic technologies. PMCloud offers a detailed platform for the synchronization of theoretical and experimental results. Based on the observed materials-based plasmonic response, the material and other parameters can be chosen as inputs, and corresponding theoretical data can be generated, directly compared, and matched with experimental data. The parameters in PMCloud can be varied interactively to find better congruence with experimentally observed data. On the other hand, the user can generate the theoretical data in PMCloud beforehand and use these as a prediction for the experimental results, and both can be cross-verified afterward. Hence, the PMCloud constitutes a unique open access synchronous platform, with which researchers from theoretical, experimental plasmonic/photonic, and corresponding sensing communities can verify their experimental and theoretical findings. Section 4.2.1 showcases a PMCloud-calculated sensing performance prediction for Au nano-spheres.

### 2.2. Material Development

Developing new materials is integral for advancing technology by taking advantage of superior material properties for different applications. For instance, better optical properties, such as an adjustable refractive index and low absorption, are driving factors for new material development in integrated optics [27]. In plasmonics, conventional materials AU and Ag have disadvantages stemming from metallic properties such as high intrinsic loss and low melting point in transformation optics, as well as photothermal and hot-electron devices, motivating a search for alternative materials [28,29,30]. Another important motivation for material development could be the cost and ease of material fabrication for optical application [31]. Furthermore, new materials for sustainable, reliable, low-cost, and efficient photocatalytic and photovoltaic applications are widely investigated [32,33,34]. In all of these applications, theoretical characterization of optical properties helps motivate and analyze material development [35]. PMCloud cloud can be a handy tool for an experimentalist to use the Drude–Lorentz model to obtain new materials’ refractive index and further investigate optical responses such as absorption, reflection, and transmission coefficients. In addition, a comparative study of existing materials, particularly in plasmonics with Au and Ag, with novel materials is also essential for plasmonic applications [36]. Section 4.1 demonstrates validated, DFT-generated refractive index spectra for low energy levels up to 4 eV, in comparison with PMcloud-generated data based on Drude–Lorentz parameters.

### 2.3. Thin Films

Thin films have various applications in optics, including solar cells, metallic glass, and plasmonics [37,38,39]. The optical characterization of thin films in solar cell applications is customary, using transmittance and absorbance spectra to calculate the optical bandgap and investigate the different fabrication parameters and film thickness [40,41,42]. Furthermore, transmittance and reflectance measurements are essential for, e.g., coating and window material applications [43], and metallic thin-film surface plasmon resonance detection, using Kretschmann configuration, is a widely used technique for high-sensitivity sensing applications [44,45]. Moreover, reflectivity and transmittance spectra are standard evaluation measures in a thin film for glass and optoelectronic applications [46]. In the PMcloud thin-film tab, quick absorption calculation for different materials, including user-created new materials and variable thickness, is provided to theoretically calculate reflection, transmission, and absorption spectra of arbitrary thin-film stacks, readily serving to verify experimental results and giving insight into the expected optical property before investing in fabrication or evaluating the fabrication method. Users can also numerically optimize thin films for the number of layers, materials, and thickness for a specific transmittance, absorptance, and reflectance. Section 4.3 showcases the PMCloud-calculated Kretschmann-configuration performance of the recently proposed plasmonic material AZO6 [47]. Additionally, Section 4.4 demonstrates the calculation of an AlAs/GaAs thin-film Bragg reflector, revealing a perfect reflectivity stop band range.

### 2.4. Teaching

In an educational setting, PMCloud’s interactive nature potentially allows for integration in lectures about photonic materials. Rather than purely literature-based teaching, students can independently explore several materials and morphological material characteristics. Each change in a material or parameter is immediately freely accessible to view online. Alongside the above-mentioned scientific use cases, students and teachers can search the database for various literature-based materials, interactively compare their optical characteristics, and even compare their performance in standard numerical experiments with varying morphological parameters without any need to install or even purchase specific software. In classes about solid-state physics, the authors envisage students who interactively create Drude and Lorentz resonances and thereby explore the effect of intra- and interband transitions on the created materials’ optical behavior.

## 3. Methods

The PMCloud is built using Python 3 [48], and it currently runs on SDU’s High-Performance Computing (HPC) facilities via the dash [49] platform. PMCloud’s backbone consists of theoretical methods to create material models and to calculate the optical response of fundamental morphologies. The following sections describe these methods and the basic workflow. PMCloud is organized in website tabs for each task, and the basic workflow is illustrated in Figure 2. Alongside this manuscript, we provide extensive Appendix A, constituting a step-by-step tutorial on using the PMCloud to generate the numerical results presented in the following sections.

### 3.1. Materials Tool

The interactive materials tool covers the optical response of the materials used in the subsequent tools. The available materials for the following tools are decided entirely by the user. It is possible to assign up to 20 material models. This list of materials can be downloaded for later use and shared with colleagues. The material models can be viewed interactively in units of the dielectric constants or complex refractive index and plotted over wavelength or energy. The various ways to generate material models are:Uploading a .txt file from experimental data, simulation, or other source.Pulling the material model from the refractive index database [22].Generating a material using Drude–Lorentz parameters.Generating a simple constant refractive index material by entering the *n* and *k* values.

#### Drude–Lorentz Material Model

The optical response of a material is a key parameter, represented in its dielectric permittivity or refractive indices. It contains useful information about the optical and electrical material nature, determining how an incoming EM wave propagates inside the material through physical phenomena such as scattering, attenuation, and velocity changes. Generating new materials with varying optical responses in wavelength/energy that are extendable over a wide range of wavelengths/energies is of significant importance for optics and photonics research, paving the way for advanced computational material and structure engineering. This section will briefly introduce the method that we utilize to generate the optical response of any arbitrary dispersive material by employing one of the most comprehensive optical models. The Drude–Lorentz model considers both classical and quantum mechanical views of electromagnetism. The optical response plot of a metal follows an inverse exponential curve at a lower energy regime, stemming from intraband transitions of electrons occupying the electronic orbitals around the Fermi level. This type of electron can easily be excited to higher energies, absorbing infinitesimal energy from incident electromagnetic waves and thus creating a significant peak in the bulk material’s optical response around zero energy. This is the classical model predicted by Paul Drude in the early twentieth century. The most crucial part—of which classical physics was unaware, and which remained unanswered until the emergence of quantum mechanics—was interband transition contributions occurring in higher-energy regimes. This mechanism comes from electron transitions between valence and conduction bands in electronic band structures, appearing in the form of individual peaks in the optical response function. The Drude–Lorentz equation for the dielectric permittivity of metals is given by:(1)$$\begin{array}{c} \epsilon \left(\omega \right)=\epsilon_{\infty }\left(\omega \right)+\sum_{j}^{}\frac{f_{j}\omega_{p}^{2}}{\left(\omega_{j}^{2}-\omega^{2}\right)-i\omega \gamma_{j}}, \end{array}$$
where $\omega_{j}$, $\gamma_{j}$, and $f_{j}$ denote the resonance frequency, the damping coefficient, and the resonance strength, respectively. $\epsilon_{\infty }$ defines the background permittivity, $\omega_{p}$ the plasma frequency, and ω the incident beam frequency. If the mentioned parameter values are known/provided, one can easily generate the respective dielectric permittivity via Equation (Equation 1).

### 3.2. Comparison Tool

The comparison tool is used to graphically compare all or some of the materials models generated using the materials tool. The materials to be compared are selected from a list of the material models and plotted interactively. Again, the material models can be examined in units of the dielectric constants or complex refractive index and plotted over wavelength or energy. The comparison tool helps compare the potentially subtle differences of various material models, experimentally obtained data, and/or library data.

### 3.3. Particle Scattering through Mie Theory

The Mie solution to Maxwell’s equations describes the scattering of an electromagnetic plane wave by geometries where it is possible to state separate equations for the radial and angular dependence of solutions, such as a homogeneous sphere [50]. Mie scattering theory approximates the scattering of particles down to sizes of approximately the wavelength of the incident light ray. However, related computational methods have enabled computation for spherical scatterers, which are a few orders of magnitudes larger than the incident wavelength [51]. This method gives us the capability to compute the scattering and absorption of spherical, coated (or non-coated) nanoparticles dispersed in a medium. The solution takes the form of an infinite series of partial waves. The extinction, scattering, and absorption efficiencies are calculated using the relations
(2)$$\begin{array}{c} Q_{ext}=\frac{2}{x^{2}}\sum_{n=1}^{\infty }\left(2n+1\right)ℜ\left(a_{n}+b_{n}\right), \end{array}$$
(3)$$\begin{array}{c} Q_{sca}=\frac{2}{x^{2}}\sum_{n=1}^{\infty }\left(2n+1\right)\left(|a_{n}{|}^{2}+|b_{n}{|}^{2}\right), \end{array}$$
(4)$$\begin{array}{c} Q_{abs}=Q_{ext}-Q_{sca}, \end{array}$$
where $x=\frac{2\pi }{\lambda }\frac{d}{2}$ with *d* as the particle diameter and λ as the EM-waves wavelength. $a_{n}$ and $b_{n}$ are the external field coefficients, calculated using the complex refractive index and the size parameter [52]. The external field coefficients, $a_{n}$ and $b_{n}$, depend on the phase angles, $\alpha_{n}$ and $\beta_{n}$, of the electric- and magnetic fields, in the following way:(5)$$\begin{array}{c} a_{n}=\frac{1}{2}\left(1-e^{-2i\alpha_{n}}\right)=i\sin\alpha_{n}e^{-i\alpha_{n}}, \end{array}$$
and
(6)$$\begin{array}{c} b_{n}=\frac{1}{2}\left(1-e^{-2i\beta_{n}}\right)=i\sin\beta_{n}e^{-i\beta_{n}}. \end{array}$$

PMCloud employs the PyMieScatt Python package [52] to compute an approximation to the infinite series of the Mie solution. So far, our code assumes that the particles and their shell are spherically symmetric and that the particles are homogeneously dispersed in a medium with inter-particle distances, such that plasmonic coupling between the particles is negligible. PMCloud allows us to calculate the scattering, absorption, and extinction cross sections interactively over the wavelength, material, and size parameter inputs, enabling us to have non/constant RI materials in all regions of the system of dispersed core–shell spherical particles in a medium.

### 3.4. Thin-Film Tool

The thin-film tool computes the optical properties of thin-film devices, i.e., reflection, absorption, and transmission of light incident on multi-layered thin-film stacks. The tool is based on the transfer matrix method (TMM) [53], which solves Maxwell’s electromagnetic wave equation by employing boundary conditions in the layer interfaces. The user generates the layers by choosing their material model based on the materials generated using the materials tool and the layer’s sequence and thickness. The sequence of layers is listed and graphically represented for the user to examine in the tool (see Figure 3).

For simplified formulation, we describe the backbone of the TMM tool using the normal incident and transverse electric field [54], as shown in Figure 3. Due to the boundary condition, the tangential electric field component *E* of the plane wave in Figure 3 is the same for both sides of the corresponding layer interface. The equality of the tangential field on the interface leads to the following relationship between the incident and the transmission region fields:(7)$$\begin{array}{c} \left(\begin{array}{c} E_{x}^{\mathrm{tr}}\\ H_{y}^{\mathrm{tr}} \end{array}\right)=M\left(\begin{array}{c} E_{x}^{\mathrm{in}}\\ H_{y}^{\mathrm{in}} \end{array}\right), \end{array}$$
where $E_{x}^{\mathrm{tr}}$ and $H_{y}^{\mathrm{tr}}$ are the electric field and the magnetic field of the transmission region. $E_{x}^{\mathrm{in}}$ and $H_{y}^{\mathrm{in}}$ are the electric and magnetic field of the incident region. *M* is a transfer matrix given by:(8)$$\begin{array}{c} M=\prod_{j=1}^{n}m_{j}, \end{array}$$
with
(9)$$\begin{array}{c} m_{j}=\left[\begin{array}{cc} \cos(k_{j}d_{j}) & \frac{1}{k}\sin\left(k_{j}d_{j}\right)\\ -k\sin(k_{j}d_{j}) & \cos(k_{j}d_{j}) \end{array}\right], \end{array}$$
where $k=\frac{n\omega }{c}$ is the wave vector of the incident field, for a refractive index *n* and speed of light *c*; $k_{j}=\frac{n_{j}\omega }{c}$ is the wave vector, and corresponds to layers for $n_{j}$, the refractive index of material layer $l_{j}$ where j=1,2,3,…,n. For normal incidence, the electric field in the incident and transmission regions is given by:(10)$$\begin{array}{c} E_{x}^{\mathrm{in}}=E\left(e^{ikz}+re^{-ikz}\right), \end{array}$$
(11)$$\begin{array}{c} E_{x}^{\mathrm{tr}}=E\left(te^{ik(z-d_{t})}\right), \end{array}$$
and
(12)$$\begin{array}{c} d_{t}=\sum_{j=1}^{n}d_{j}, \end{array}$$
where *r* and *t* are the reflection and transmission coefficient, respectively. Using Equations (Equation 10) and (Equation 11), we can solve for the transmission and reflection coefficients using the simplified Equation (Equation 13). The absorption coefficient can be calculated by adding the transmission and reflection coefficients
(13)$$\begin{array}{c} \left(\begin{array}{c} t\\ ikt \end{array}\right)=M\left(\begin{array}{c} 1+r\\ ik(1-r) \end{array}\right). \end{array}$$

Next, the type of calculation is specified—the user can choose to sweep over incidence light wavelength (energy), incidence light angle, or both simultaneously. Depending on the type of calculation, the user must specify the wavelength range, incidence angle range, and sampling rates.

The simulation results are plotted interactively, and the user can download the sequence of layers for later use or share the sequence with a colleague. The user can also edit this list of layers using any text editor—this is currently recommended if the user wishes to simulate a system of recurring layers, such as a Bragg reflector. Consult the Appendix A for a step-by-step tutorial explaining the process.

## 4. Numerical Results and Discussion

This section showcases PMCloud’s capability of creating state-of-the-art research results within the previously mentioned use cases and beyond.

### 4.1. Materials Tool: Drude–Lorentz Parameters

Starting with the materials tool, we can demonstrate that Drude–Lorentz parameters to generate a material model are accurate. We use the findings of two papers [47,55], where we have computationally investigated the plasmonic behavior of Zirconium Nitride (ZrN) and Aluminum-doped Zinc Oxide (AZO), using density functional theory (DFT) and electromagnetic modeling to compare the refractive indices of these materials. The mentioned papers use DFT calculations for the material model in refractive indices (dielectric permittivity), along with Drude–Lorentz model calculations, to extract optical parameters. Here, we enter these parameters into the materials tool’s Drude–Lorentz tab. The Drude–Lorentz parameters for two different percentages of Al, doped into ZnO, demonstrated by AZO6, AZO2, as well as ZrN, are listed in Table 1, Table 2 and Table 3, respectively. Each resonance indicates a Lorentzian peak in refractive indices with a center of resonance at $\omega_{j}$, broadening factor $\gamma_{j}$, and peak strength of $f_{j}$. The first resonance in each table corresponds to the Drude model, bringing metallic behavior to the material. The remaining peaks are associated with dielectric material properties, originating from the excitation of electrons between electronic orbitals (Lorentz terms). The DFT-calculated plasmon frequencies read 1.515, 2.193, and 7.456 eV for AZO2, AZO6, and ZrN, respectively.

To further investigate the accuracy of our implemented method for the material model, we have depicted the refractive indices of DFT calculations together with the ones coming from the Drude–Lorentz tab for listed materials and respective optical parameters. The results of the comparison for ZrN material models can be seen in Figure 4 in the wavelength range 300–1200 nm. One can see that the material model in the cloud tab matches correctly with DFT results. They both show that the metallic behavior of ZrN appears in the early visible spectrum (≈400 nm), confirming the intermetallic nature and resembling the most famous cases such as gold, silver, and TiN. It is also visible that the interband transitions occur for the wavelength below 400 nm.

Similarly, for Aluminum-doped Zinc Oxide (AZO), we compare the DFT-generated refractive index to the index obtained using PMCloud’s materials tool (Drude–Lorentz input). The comparison of AZO2 and AZO6 can be seen in Figure 5 and Figure 6. One can see that two proposed materials demonstrate metallic behavior at different optical regimes compared to the ZrN case. Their metallic properties lie in a near-infrared regime (NIR) with an epsilon near zero feature between interband and intraband transitions coming from the wide bandgap characteristics of the parent compound ZnO. These two features, metallic properties together with epsilon-near-zero behavior, make AZO an interesting candidate for future meta-material applications in a telecommunication regime. It is also seen that increasing Al dopant shifts further metallic behavior to the visible window in case of AZO6, while the interband transitions also shift to lower wavelengths.

We observe that the method of entering Drude–Lorentz parameters into the materials tool works very well, since the DL curves in the figures lie very close to the DFT curves and exhibit all the same features. We can conclude that our web-based material model cloud is a reliable tool for generating any new material based on the Drude–Lorentz model. Since it demonstrates condensed-matter physics concepts such as electronic transitions between atomic orbitals and energy band structures, we can find lots of information about its physics while inserting every Lorentzian peak separately to define a new material. The Appendix A describes in detail how to reproduce the RI curves using the DL parameters method.

### 4.2. Particle Scattering

#### 4.2.1. Light Scattering from a Gold Sphere for Sensing Applications

We demonstrate the capability of the particle scattering tool to assist the prediction of biomedical sensing with gold nanoparticles. Figure 7 demonstrates how PMCloud allows us to interactively create extinction spectra for gold spheres, for varying surrounding medium (analyte) index, and for varying sphere diameters. This way, the researcher can virtually experiment with finding the right nanoparticle material, size, and (spherical) composition, and predict the resonance shift.

#### 4.2.2. Light Scattering from AZO and ZrN Spheres

We demonstrate the capabilities of the particle scattering tool by using the generated Drude–Lorentz material models as particle cores in this section.

The Scattering efficiencies are computed for spherical particle cores with 65nm diameter, dispersed in water (n=1.33,k=0). We compare particle core materials of AZO6, AZO2, and ZrN with the same particle diameter and dispersed in the same medium (water). The materials are generated using the materials clouds’ method of generating Drude–Lorentz materials, as described in the previous section. The comparison is plotted in Figure 8.

PMCloud can interactively compute the scattering efficiencies of the various particle materials, and we observe that the scattering efficiency peaks vary widely depending on the core material. For this specific system of dispersed particles, the core of the particles consisting of ZrN material peaks at λ=530 nm, -AZO6 at λ=1255 nm and -AZO2 at λ=1985 nm. These peaks are associated with localized surface plasmon phenomena originating from the interaction of incident light with conduction electrons at the metallic nanoparticle–dielectric interface. As it is known, depending on the materials in use and the size of nanoparticles, one may observe different orders of modes excited on the surface of nanoparticles, such as electric dipole and quadrupole. In our case, only electric dipole mode is excited in all three materials. In the case of excitation of higher-order plasmonic modes, one needs to increase the size of nanoparticles. As expected before, the peak position for the particle with ZrN as the core material has a peak in the visible spectrum, resembling gold and silver, and is attractive for plasmonics applications within the visible light spectrum. In contrast, AZO materials are more applicable in the infrared region for telecom applications. AZO2, the case with fewer Al dopants (≈2%), shows plasmonic properties at the NIR regime, whereas the plasmonic peak of AZO6 blue-shifted due to the larger population of conduction electrons resulting from more Al dopant. These results confirm the findings of our previous papers, in which we discussed the potential plasmonic behavior in meta-materials made of ZrN and AZO compounds [47,55].

One of the main advantages of the particle scattering tool is that it can tweak the wavelength position of the peaks near the wavelength region not only by adjusting the particle diameter, but also by adding a particle shell to make a core–shell scattering system. Furthermore, as the shell’s impact has been intensively proven in a core–shell system, the particle scattering tool gives us the opportunity to quickly investigate the effect of varying the shell material and thickness and trying different materials for the surrounding medium from the materials clouds.

### 4.3. Kretschmann Configuration for AZO6

Plasmon resonance-based excitation via the Kretschmann configuration is a commonly used method in thin-film-based refractive index sensing. Usually, an incident electromagnetic wave enters a medium through a dielectric prism to observe the electromagnetic loss [57]. The incident angle (Kretschmann angle) and the wavelength in which a minimum reflection occurs due to the complex dielectric permittivity of the medium confirms surface plasmon resonance. In this demonstration, we used a polychromatic incident wave, with 1600–2000 nm wavelength and an incident angle in the range of 55–90°, to generate a heat-map for the AZO6 absorption efficiency. From Figure 9, we observe a sharp increase in absorption in the wavelength range of 1700–2000 nm and a 70–80° incident angle, corresponding to the plasmonic resonance of AZO6. The Kretschmann angle for AZO6 varies depending on the wavelength used to excite the system.

### 4.4. An AlAs/GaAs Thin-Film Bragg Reflector

One of the fascinating thin-film phenomena is the so-called Bragg reflector, a stratified medium usually built from multiple alternating dielectric layers of varying refractive indexes or layer thicknesses [58,59,60]. The resulting periodic variation in the effective refractive index leads to superimposed partial reflection in the incidence region and potentially to constructive interference (in reflection). This interference, if engineered correctly, can act as a high-quality reflector for, e.g., waveguiding and laser-cavity applications [61,62]. The high-reflectance wavelength spectrum range (stop band) will increase as the difference between the high and low refractive index increases. The number of alternating layers also affects the reflectance; as the number of alternating layers increases, so does the reflectance [63].

A classical thin-film Bragg reflector design consists of a so-called *quarter-wave stack* of alternating high- and low-index layers, where all layers are of the same optical path length $t_{h}n_{h}=t_{l}n_{l}=\lambda_{\mathrm{center}}/4$, with the intended photonic stop band’s center wavelength $\lambda_{\mathrm{center}}$ [64,65]. The corresponding reflectance *R* at the center wavelength $\lambda_{\mathrm{center}}$, under normal incidence, is approximately given by
(14)$$\begin{array}{c} R={\left(\frac{1-\left(\frac{n_{s}}{n_{a}}\right){\left(\frac{n_{h}}{n_{l}}\right)}^{2N}}{1+\left(\frac{n_{s}}{n_{a}}\right){\left(\frac{n_{h}}{n_{l}}\right)}^{2N}}\right)}^{2}, \end{array}$$
where $n_{s}$ is refractive index of the substrate, $n_{a}$ is the refractive index of the incident (ambient) region, and *N* is the number of repeating layers. It is immediately obvious from Equation (Equation 14) that, under usual conditions ($n_{h}>n_{l}$), *R* tends to 1 (perfect reflection) for N→∞. Furthermore, one can approximate the stop band’s frequency width Δf with respect to its center frequency $f_{\mathrm{center}}$ by [66]
(15)$$\begin{array}{c} \frac{\Delta f}{f_{\mathrm{center}}}=\frac{4}{\pi }\arcsin\left|\frac{n_{l}-n_{h}}{n_{l}+n_{h}}\right|. \end{array}$$

It is, however, less obvious how the system (i.e., reflectivity and stop band width) reacts to non-normal incidence angles. Here, we calculate the reflection dispersion (spectrum vs. incidence angle) for a quarter-wave stack made of gallium arsenide (GaAs) and aluminum arsenide (AlAs), for an intended stop band center wavelength $\lambda_{\mathrm{center}}=1000\;\mathrm{nm}$. At this wavelength, the refractive indices of GaAs and AlAs are approximately $n_{h}=3.49$ and $n_{l}=2.95$, respectively [67]. Consequently, the corresponding layer thicknesses read $t_{h}=71.6\;\mathrm{nm}$ and $t_{l}=84.7\;\mathrm{nm}$, respectively.

Figure 10 shows the reflectivity of the Bragg reflector created using 20 alternating layers of the above-mentioned dimensions, calculated by the thin-film tool contained in the PMCloud. We can observe a perfect reflection stop band from 930 nm–1030 nm for all incident angles. The thin-film tool is instrumental in designing Bragg reflectors, providing a platform for calculating the reflection, transmission, and, consequently, bulk absorption for various layer-sequence, material, wavelength, and incident angle combinations. We provide a complete demonstration for creating Figure 10 using the thin-film tool in the Appendix A.

## 5. Conclusions and Future Plans

It is a lengthy process for students and researchers to acquire a basic familiarity with fundamental photonics effects, such as the plasmonic effect and its parameters. In the present work, we demonstrated the use of PMCloud as an interactive educational tool for optical material properties and photonic phenomena, including material models, comparisons of these, particle scattering, and thin-film optical response. We employed PMCloud to show a novel plasmonic property comparison of newly proposed ZrN and AZO materials, illustrated its capability of desinging and optimizing multi-layer thin-film stacks, and demonstrated its ability to interactively investigate metal nanoparticle-based refractive index sensing. We envision the PMCloud as a growing learning and design platform, which already contains modules ranging from fundamental materials models to the simulation of plasmonic responses for spherical particles and thin-film stacks. Shortly, PMCloud will be further extended, including corrugated thin-film structures, non-spherical particles, and more. The authors welcome collaborations with researchers and educators in the field, aiming to provide effective and efficient tools in various aspects of the area for a global audience.

## Figures and Tables

**Figure 1 nanomaterials-12-02585-f001:**
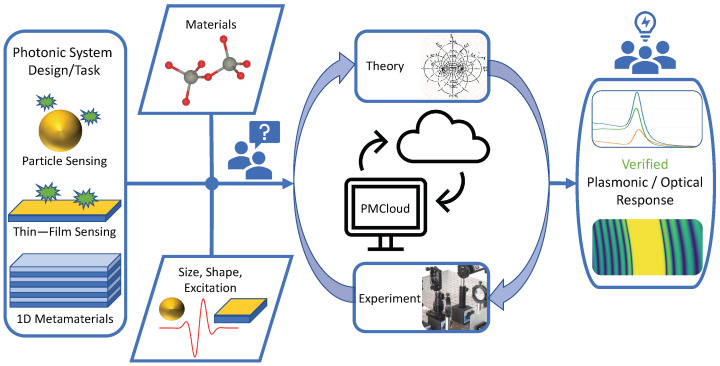
The Photonic Materials Cloud (PMCloud) is an easy-access platform used to design or cross-validate experimental setups and results. It allows the upload of experimental material data and a direct comparison to database- or theory-based materials. With the uploaded or generated data, the user can perform a series of standard scattering and thin-film optical experiments, predicting the optical response of various geometries, thereby helping to validate or modify experiments time efficiently.

**Figure 2 nanomaterials-12-02585-f002:**
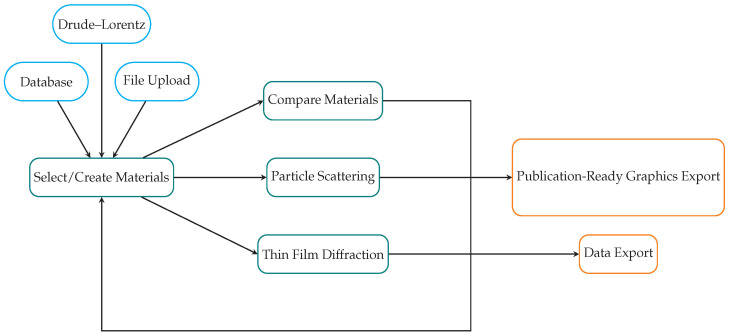
PMCloud’s basic workflow. Blue indicates possible input channels, green represents the current analysis “tabs”, and orange marks possible export channels.

**Figure 3 nanomaterials-12-02585-f003:**
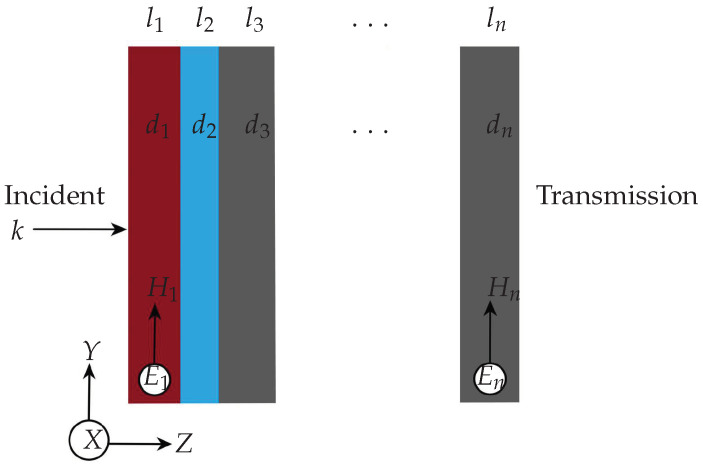
Schematic diagram for 1D multi-layered thin-film structure where magnetic field lies on incident plane, a plane wave propagating in wave vector *k* direction with layer thicknesses $d_{i}$ for layers $l_{i}$, i∈{1,…,n}.

**Figure 4 nanomaterials-12-02585-f004:**
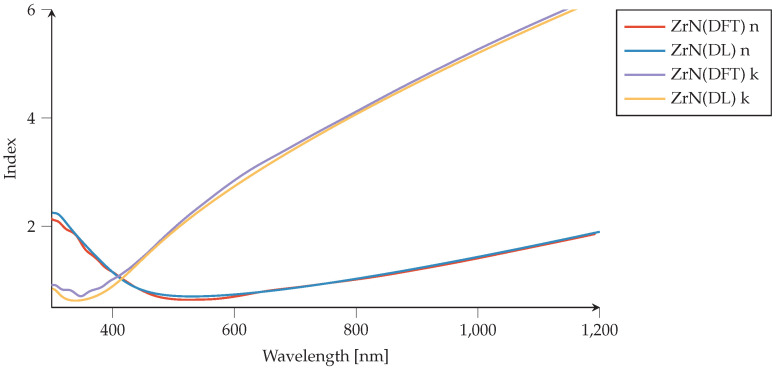
Zirconium nitride (ZrN) comparison of refractive indices (*n*, *k*). Material models generated via density-functional theory (DFT) [55] are compared to the ZrN material model generated with Drude–Lorentz (DL) parameters using PMCloud’s materials tool. We use $\epsilon_{\infty }=0.6$.

**Figure 5 nanomaterials-12-02585-f005:**
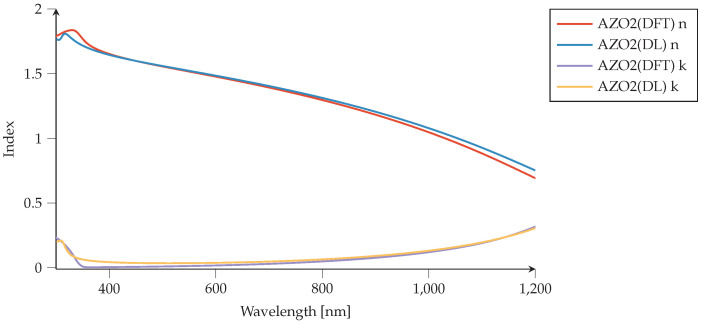
Aluminum-doped zinc oxide (AZO2) comparison of refractive indices (nk). Material models generated via density-functional theory (DFT) [47] are compared to the AZO2 material model generated by Drude–Lorentz (DL) parameters using PMCloud’s materials tool. We use $\epsilon_{\infty }=1.9$.

**Figure 6 nanomaterials-12-02585-f006:**
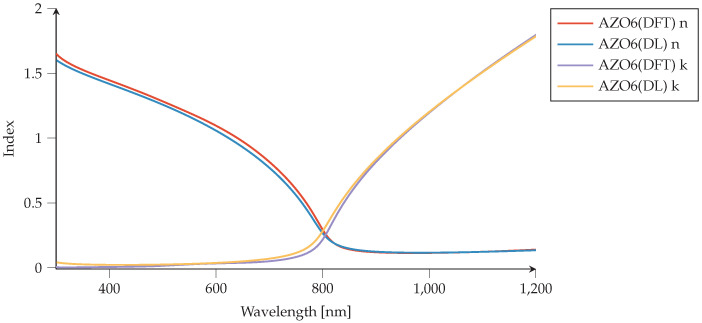
Aluminum-doped zinc oxide (AZO6) comparison of refractive indices (*n*,*k*). Material model generated via density-functional theory (DFT) [47] are compared to the AZO6 material model generated by Drude–Lorentz (DL) parameters using the Plasmonics Materials Cloud’s materials tool. We use $\epsilon_{\infty }=1.8$.

**Figure 7 nanomaterials-12-02585-f007:**
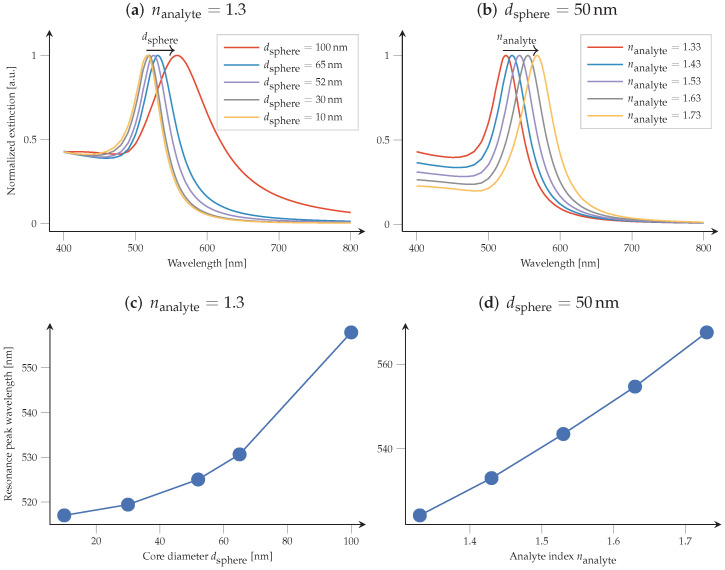
Visualising the parameter variation effect of gold spheres dispersed in an analyte. Extinction resonance shift for (**a**): changing sphere diameter, dispersed in water ($n_{\mathrm{analyte}}=1.33$), and (**b**): for constant-diameter ($d_{\mathrm{sphere}}=50\;\mathrm{nm}$) spheres, and varying analyte index $n_{\mathrm{analyte}}$. (**c**,**d**) showcase the extracted resonance peak wavelengths for the experiments shown in (**a**,**b**), respectively.

**Figure 8 nanomaterials-12-02585-f008:**
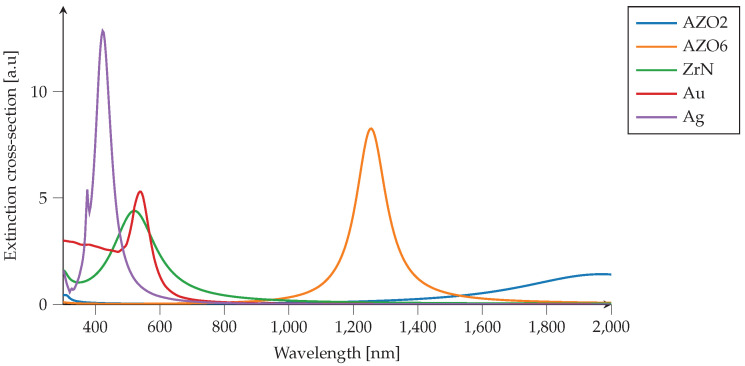
Comparison of particle scattering for spherical nanoparticles with 65nm diameter and dispersed in water with n=1.33, k=0. The particle cores are AZO6, AZO2, ZrN, Au [56] and Ag [56], which are generated from materials library for noble metals and using PMCloud’s method of generating Drude–Lorentz materials, described in the previous section. Computed using the materials clouds’ particle scattering tool.

**Figure 9 nanomaterials-12-02585-f009:**
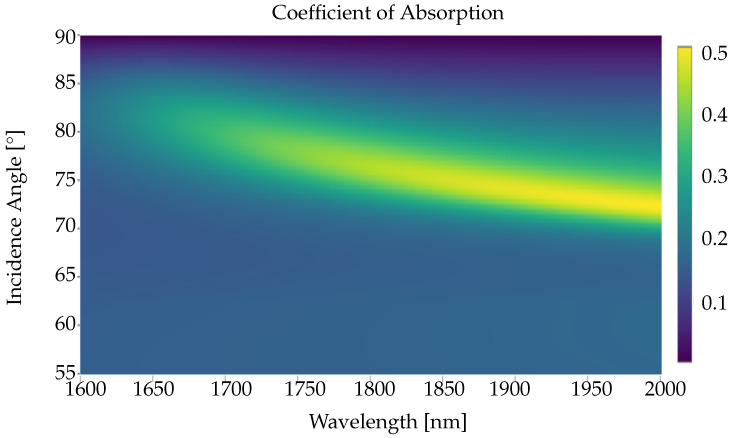
Transfer matrix method calculated absorption efficiencies for a 200 nm-thick AZO6 thin film, sandwiched between silicon dioxide (SiO_2_; n=1.52) and air (n=1). A poly-chromatic plane wave is incident through the SiO_2_ region, polarized within the plane of incidence (TM).

**Figure 10 nanomaterials-12-02585-f010:**
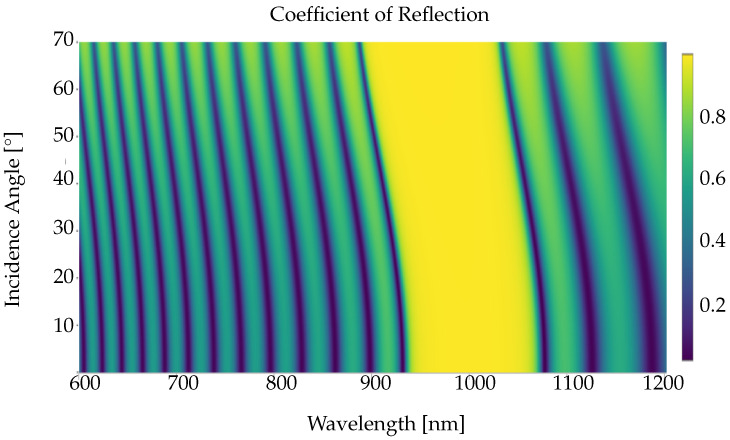
Reflection dispersion (reflectivity vs. spectrum and incidence angle) for a quarter-wave stack (Bragg mirror) made of 20 alternating gallium arsenide (GaAs) and aluminum arsenide (AlAs) films, with refractive indices $n_{h}=3.49$ and $n_{l}=2.95$, respectively, and layer thicknesses $t_{h}=71.6\;\mathrm{nm}$ and $t_{l}=84.7\;\mathrm{nm}$, respectively. Our calculations reveal an angle-robust stop band between 930 nm and 1030 nm incident wavelength.

**Table 1 nanomaterials-12-02585-t001:** The optical parameters of Drude–Lorentz model for AZO6 obtained from DFT-GGA calculations [47].

Resonance *j*	$\omega_{j}$ [eV]	$\gamma_{j}$ [eV]	$f_{j}$
1	0	0.05	1
2	0.294	0.213	0.243
3	4.810	0.286	0.186
4	5.258	0.433	0.351
5	5.863	0.613	0.597
6	6.607	0.741	0.890
7	7.449	0.816	1.237
8	8.159	0.772	1.281
9	8.790	0.791	2.728

**Table 2 nanomaterials-12-02585-t002:** The optical parameters of Drude–Lorentz model for AZO2 obtained from DFT-GGA calculations [47].

Resonance *j*	$\omega_{j}$ [eV]	$\gamma_{j}$ [eV]	$f_{j}$
1	0	0.2	1
2	0.269	0.216	0.088
3	3.986	0.240	0.148
4	4.280	0.352	0.284
5	4.634	0.527	0.519
6	5.098	0.742	0.819
7	5.712	0.998	1.204
8	6.489	1.317	1.882
9	7.495	1.724	3.508
10	8.879	1.697	12.81

**Table 3 nanomaterials-12-02585-t003:** The optical parameters of Drude–Lorentz model for ZrN obtained from DFT-GGA calculations [55].

Resonance *j*	$\omega_{j}$ [eV]	$\gamma_{j}$ [eV]	$f_{j}$
1	0	0.62	1
2	0.18	0.24	0.03
3	4.06	0.35	0.02
4	4.75	0.76	0.17
5	5.29	1.26	0.48
6	5.72	0.45	0.09
7	6.62	2.27	1.34
8	7.25	0.40	0.12
9	8.01	1.05	0.55

## Data Availability

The data displayed in this manuscript are generated and can be freely recreated using PMCloud [21].

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
