# Peer review of "Photonic Materials Cloud: An Online Interactive Open Tool for Creating, Comparing, and Testing Photonic Materials"

_nanomaterials, 2022, doi:10.3390/nano12152585_

Round 1

Reviewer 2 Report

This work provide An Online Interactive Open Tool for Creating, Comparing, and Testing Photonic Materials. And show some practical instance to demonstate its feasibility. It seems with valuable contribution to the commity. However, before the recommendation, some concerns should be addressed.

1)  There are three critical function provided in this platform, material modeling, particle plasmonic and thin film diffraction. And these three functions all have corresponding commercial tools. However, the paper should highlight the difference between the commercial softwere and the tool provided in this work

2)  Noting the motivation of this Interactive platform is not only for research but also for education. However, there is not sufficient contents to point out the special Interactive design for, e.g., teaching purpose.

3)   Ref S2, the website http://photonicmaterials.eu. Is not able to explored normally. This is not in line with the purpose as an Photonic Materials Cloud. The link should be better-maitained.  

Round 2

Reviewer 1 Report

The authors have revised the manuscript carefullly. Now I recommend the acceptance of this manuscript.